# Melanoma Detection Using Deep Learning-Based Classifications

**DOI:** 10.3390/healthcare10122481

**Published:** 2022-12-08

**Authors:** Ghadah Alwakid, Walaa Gouda, Mamoona Humayun, Najm Us Sama

**Affiliations:** 1Department of Computer Science, College of Computer and Information Sciences, Jouf University, Sakaka 72341, Al Jouf, Saudi Arabia; 2Department of Computer Engineering and Networks, College of Computer and Information Sciences, Jouf University, Sakaka 72341, Al Jouf, Saudi Arabia; 3Department of Information Systems, College of Computer and Information Sciences, Jouf University, Sakaka 72341, Al Jouf, Saudi Arabia; 4Faculty of Computer Science and Information Technology, Universiti Malaysia Sarawak, Kota Samarahan 94300, Sarawak, Malaysia

**Keywords:** deep learning, machine learning, convolutional neural network, HAM10000, skin lesion, ESRGAN

## Abstract

One of the most prevalent cancers worldwide is skin cancer, and it is becoming more common as the population ages. As a general rule, the earlier skin cancer can be diagnosed, the better. As a result of the success of deep learning (DL) algorithms in other industries, there has been a substantial increase in automated diagnosis systems in healthcare. This work proposes DL as a method for extracting a lesion zone with precision. First, the image is enhanced using Enhanced Super-Resolution Generative Adversarial Networks (ESRGAN) to improve the image’s quality. Then, segmentation is used to segment Regions of Interest (ROI) from the full image. We employed data augmentation to rectify the data disparity. The image is then analyzed with a convolutional neural network (CNN) and a modified version of Resnet-50 to classify skin lesions. This analysis utilized an unequal sample of seven kinds of skin cancer from the HAM10000 dataset. With an accuracy of 0.86, a precision of 0.84, a recall of 0.86, and an F-score of 0.86, the proposed CNN-based Model outperformed the earlier study’s results by a significant margin. The study culminates with an improved automated method for diagnosing skin cancer that benefits medical professionals and patients.

## 1. Introduction

Cells that proliferate and divide uncontrollably are referred to as “cancer”; they can quickly spread and invade nearby tissues if left untreated. Any sort of cancer, not just skin cancer, has the most significant probability of developing into a malignant tumor [1,2]. Melanoma (mel), Basal-cell carcinoma (BCC), nonmelanoma skin cancer (NMSC), and squamous-cell carcinoma (SCC) are the most common forms of skin cancer. It should be noted that some kinds of skin cancer, such as actinic keratosis (akiec), Kaposi sarcoma (KS), and sun keratosis (SK) are scarce [3]. Skin cancer of all varieties is increasing, as illustrated in Figure 1.

Malignant and non-malignant skin cancers are the most common [1,5]. The presence of cancerous lesions exacerbates cancer morbidity and healthcare expenses. Consequently, scientists have focused their efforts on creating algorithms that are both highly precise and flexible when it comes to spotting early signs of cancer in the skin. Malignant melanocyte cells proliferate, invade, and disseminate rapidly; therefore, early detection is critical [6]. Dermoscopy and epiluminescence microscopy (ELM) are frequently used by specialists to identify if a skin lesion is benign or cancerous.

A magnifying lens and light in dermatology are used to see medical patterns better such as hues, veils, pigmented nets, globs, and ramifications [7,8].. Visually impaired people can see the morphological structures that are otherwise hidden. These include the ABCD (Asymmetrical form, Border anomaly, Color discrepancy, Diameter, and Evolution) [9], 7-point checklist [10], and pattern analysis [11]. Non-professional dermoscopic images have a predictive value of 75% to 80% for Melanoma, but the interpretation takes time and is highly subjective, depending on the experience of the dermatologist [12]. Computer-Aided Diagnosis (CAD) approaches have made it easier to overcome these difficulties [8,12]. CAD of malignancies made a giant leap forward thanks to Deep Learning (DL)-based Artificial Intelligence (AI) [13,14]. In rural areas, dermatologists and labs are in poor supply; therefore, using DL approaches to classify skin lesions could help automate skin cancer screening and early detection [15,16]. To classify images in the past, dermoscopic images strongly depended on the extraction of handcrafted characteristics [17,18]. Throughout these promising scientific advances, the actual deployment of DCNN-based dermoscopic pictures has yielded amazing results. Still, future development of diagnosis accuracy is hampered by various obstacles, such as inadequate training data and imbalanced datasets, especially for rare and comparable lesion types. Regardless of the restrictions of the dataset, it is vital to maximize the performance of DCNNs for the correct Classification of skin lesions [14,19].

Models such as CNN, and modified Resnet50, are used in this research. We found that the invented CNN model beats existing DCNNs in classification accuracy while testing their performance on the HAM10000 dataset. To select the best network for diverse medical imaging datasets, it may be necessary to conduct multiple experiments. Accordingly, the paper’s primary contributions can be summarized in this way:We used an enhanced generative adversarial network with super-high resolution (ESRGAN) with 10,000 training photos to produce high-quality images for the Human Against Machine dataset (HAM10000 dataset [20]) to enhance the visibility of the images. ERSGAN improves the accuracy of Classification.Segmentation is performed for each image in the dataset to specify ROI to facilitate the learning process.We used Augmentation to ensure that the HAM10000 dataset had an even distribution of data.The feasibility of the suggested system is determined by a thorough comparative evaluation using numerous assessment measurements, such as accuracy, recall, precision, confusion matrix, top 1 accuracy, top 2 accuracy, and the F-score.Pre-trained networks’ weights are fine-tuned with the help of the HAM10000 dataset and a modified version of Resnet-50.The recommended technique’s overall effectiveness has been enhanced due to this change. Overfitting is prevented by using an alternative training process supported by applying various training strategies (e.g., batch size, learning rate, validation patience, and data augmentation).

This study provides an optimization strategy incorporating a CNN model a transfer learning model for detecting multiple skin lesions. Additionally, we utilized a revised form of Resnet-50 to train the weights of each Model before using it. Comparing the models’ output using images of skin lesions from the HAM10000 dataset is necessary. The dataset has a class imbalance, necessitating an oversampling approach. The paper will proceed in accordance with this arrangement. Section 2 describes the relevant research work; after that, Section 3 illustrates the dataset and the proposed approach. The following Section 4 provides and analyzes the outcomes of the suggested technique described in Section 3; this study concludes with Section 5.

## 2. Related Work

The development of a CAD procedure for skin cancer has been the basis of several investigations [21,22]. CAD systems have followed the standard medical image analysis pipeline using classical machine learning approaches for skin lesion image processing [21]. In this pipeline, image preparation, fragmentation, extraction of features, and classifications have all been tried numerous times with little success. In skin cancer research, image processing, machine learning, CNN, and DL have all been used [23] in the past. Traditional image identification algorithms necessitate feature estimate and extraction, whereas deep learning can automatically exploit the images’ deep nonlinear relationships [24,25]. CNN was the first DL model employed for skin lesion image processing. Some of the most recent deep learning studies are summarized in the following lines.

For instance, Haenssle et al. [26] analyzed a Google Inception V4 deep learning model against 58 dermatologists’ diagnoses. The data collection includes one hundred patients’ images (dermoscopic and digitalized) and medical records. Additional research presented by Albahar [24] generated an improved DL model for detecting malignant Melanoma. Model results were compared to dermatologist diagnoses from 12 German hospitals, where 145 dermatologists used the Model to arrive at their conclusions. Li et al. [27] reviewed CNN deep learning models with 99.5 percent of the time; residual learning and separable convolution are the greatest methods for constructing the most accurate Model. This level of precision, however, was only possible since the problem was binary in nature. 

For automated Diagnosis, Pacheco et al. [25] developed a smartphone app that used images of skin lesions and clinical data to identify them. The study looked at the skin lesions of 1641 persons with six types of cancer. An experimental three-layer convolutional neural network, GoogleNet, ResNet, VGGNet, and MobileNet, was compared by researchers. Initially, images of lesions taken with smartphones were used as teaching aids, but later, images of both sorts of lesions were included (clinical descriptions and images of skin lesions). The original Model’s accuracy was 0.69 percent, but clinical data increased that to 0.764 percent. To improve upon Pacheco’s findings, a new study was proposed. Based on dermal cell images, a model-driven framework for melanoma diagnosis was created by Kadampur and Riyaee [27]. With the help of the HAM10000 dataset, several deep-learning models attained an area under the curve (AUC) of 0.99. To categorize malignant and benign skin lesions, two CNN models were employed by Jinnai et al. [28]. Results of the Model were compared to dermatologists’ diagnoses and found to have a superior classification accuracy than dermatologists, according to the results.

Furthermore, Prassanna et al. [29] proposed a deep learning-based system for high-level skin lesion segmentation and malignancy detection by building a neural network. It accurately recognizes the edge of a significant lesion and designs a mobile phone model using deep neural network transfer learning and fine-tuning to improve prediction accuracy. Another approach presented by Panja et al. [30] classifies skin cancer as melanoma or benign; feature extraction was used to retrieve damaged skin cell features using a CNN model after segmenting skin images. In [31], researchers classify ISIC 2019 Dataset photos into eight classes. ResNet-50 was used to train the Model by evaluating initial parameter values and altering them using transfer training. Images outside these eight classifications are classified as unknown.

Skin cancer detection relied heavily on the transfer learning idea. According to Kassem et al. [32], a study utilizing the GoogleNet pre-trained model for eight categories of skin cancer lesions produced an accuracy of 0.949. This time, a dermatoscope, a medical device used to examine skin lesions, was used to test the proposed YOLOv2-SquuezeNet’s segmentation and drawback performance. Using the equipment considerably improved the capacity to make an early diagnosis. Table 1 shows that several deep-learning models have been implemented to categorize skin cancer in current history.

## 3. Research Methodology

The authors of this study developed a smart classification algorithm and an automated skin lesion segmentation based on dermoscopic images. We used Resnet-50 and a CNN to perform machine learning in this case.

### 3.1. Dataset Overview

Skin Cancer MNIST: HAM10000 [20] provided the benchmark datasets used in this investigation. The CC-BY-NC-SA-4.0 licensed dataset is a reliable source of information for skin cancer diagnosis. Kaggle’s public Imaging Archive was used to gather the data. A total of 10,015 JPEG skin cancer training images from two locations, one in Vienna, Austria, and the other in Queensland, Australia, were just compiled into a single dataset for training purposes. The Australian site used PowerPoint files and Excel databases to hold images and metadata. The Austrian site started collecting images with pre-digital cameras and preserved them in several formats. Based on the research, a variety of approaches are endorsed [31,32,33,34,35,36,37,38,39,40]. Using data from this benchmark, the Resnet-50 and the suggested CNN are trained to identify skin cancer in this study. In this dataset, all of the essential diagnostic categories for pigmented lesions were included, such as: akiec, benign keratosis-like lesions (bkl), bcc, dermatofibroma (df), melanocytic nevi (nv), mel, and vascular lesions (vasc). The HAM10000 dataset is presented in the illustrative form Figure 2.

### 3.2. Proposed Methodology

Figure 3 depicts the overall process of the suggested method, based on the dataset mentioned in this article, which was used to develop an automatic skin lesion classification model. Dermoscopic skin lesion images are utilized to aid in the Classification of skin cancer, and the proposed Model’s entire operational approach displays the functional architecture of that module. After preprocessing, Classification and Resnet-50/CNN-based training are the primary steps in the given Model’s functioning. ESRGAN is used to perform the initial preprocessing step, which includes image quality improvement. Ground truth images are then used to determine an augmented image’s region of interest (ROI) for each segmented lesion. Lastly, the dermoscopy image is sent to the Resnet-50/CNN models for instantaneous skin lesion and smart classification training and exposure. An intelligent classification model and an automated procedure for segmenting skin lesions are used to create the following sections of the research study, which are described in depth on each stage in the process.

#### 3.2.1. ESRGAN Preprocessing

It was important to improve the quality of dermoscopic images and eliminate multiple kinds of noise from skin lesion images in order to carry out the proposed strategy. Ensuring that the image is as clear as possible is critical to creating a reliable skin lesion categorization model. In this step, first, we perform ESRGAN to improve the overall accuracy of the image; after that, Augmentation of data is used to overcome the problem of class unbalanced; then, all the images are resized to 224 × 224 × 3, and, finally, normalization is performed.

SRGAN [43], Enhanced SRGAN, and other approaches can help improve skin lesions. The Enhanced Super Resolution GAN is an improved version of the Super Resolution GAN [44]. Regarding Model micro gradients, a Convolution trunk or a basic Residual Network is unnecessary. In addition, there is no batch normalization layer in the Model to smooth out the image. As a result, ESRGAN images can better resemble the sharp edges of image artifacts. ESRGAN employs a Relativistic Discriminator to decide whether an image is true or false [45]. The results are more accurate using this strategy. Relativistic Average Loss and Pixelwise Absolute Difference are used as loss functions in training data. The abilities of the generator can be honed through a two-stage training process. Local minima can be avoided by reducing the pixelwise L_1_ distance among the source and targeting high images. Second, the smallest artifacts are to be improved and refined in the second step. It is interpolated between the adversarially trained models and the L_1_ loss for a photo-realistic reconstruction of the original scene.

In order to discriminate between super-resolved images and genuine photo images, a discriminator network was trained. Lesion images were improved by rearranging brightness values in the histogram of the original image using an adaptive contrast enhancement technique. As a result, the procedure in Figure 4 improves the appearance of the picture’s borders and arcs while also raising the image’s contrast.

#### 3.2.2. Segmentation

Following the protocol for preparing images, ROI from the dermoscopy image is segmented. To generate ROI in each image, a ground truth mask, which was provided by the HAM10000 dataset for general purpose usage, would be applied to the enhanced image, as demonstrated in Figure 5.

#### 3.2.3. Data Augmentation

We performed data augmentation on the training set before exposing the deep neural network to the original dataset images in order to boost the dataset’s image number and address the issue of an imbalanced dataset. Adding more training data to deep learning models improves their overall performance. We can use the nature of dermatological images to apply many alterations to each image. The deep neural network does not suffer if the image is magnified, flipped horizontally/vertically, or rotated in a specific number of degrees. Regularizing the data and reducing overfitting are two goals of data augmentation, as well as addressing the dataset imbalance issue. The horizontal shift augmentation is one of the transformations used in this study; it adjusts the image pixels horizontally while maintaining the image dimension using an integer between zero and one indicating the step size for this process. Rotation is another transformation; a rotation angle between 0 and 180 is selected, and then the image is rotated randomly. The images were resized with a zoom range of 0.1, a rescale of 1.0/255, and a recommended input size of 244 × 244 × 3. In order to generate new samples for the network, all previous modifications are applied to the training set’s images. Figure 6 demonstrates how adding slightly changed copies of either current data or new synthetic data produced from the existing data is the primary goal of data augmentation.

Using data augmentation approaches, researchers can overcome the problem of inconsistent sample sizes and complex classifications. This dataset, the HAM dataset, clearly illustrates the term “imbalanced class”, which refers to the unequal distribution of samples across distinct classes, as described in Table 2 and Figure 7. Following the augmentation approaches, the new dataset is shown in Figure 8. The classes are clearly balanced after using augmentation techniques on the dataset.

#### 3.2.4. Learning Models

This section describes the basic theory of the adopted approaches, and the proposed DL approach is presented in the next sections.

##### Model Training Using CNN

Dermoscopic images of a single skin lesion were utilized for training the Model using a CNN classifier. A suitable input set for CNN is made up of many skin cancers, such as melanomas and nonmelanomas, basal cell carcinomas and squamous cell carcinomas and Melanoma, Merkel cell carcinomas and cutaneous T-cell lymphomas, as well as Kaposi sarcoma.

As depicted in Figure 9, the proposed CNN architecture includes a proposed classification model that aids in strengthening the accuracy of the proposed mechanism’s Classification. In terms of artificial neural networks, CNNs are the most advanced, thanks to their deep architectures-based architecture (ANNs). It was not until 1989 that LeCun et al. [46,47] presented the notion of CNN, an enhanced and more complex version of ANN with a deep architectural structure, as presented in Figure 9. The segmented ROIs are sent as source data to the convolutional layer of CNN when they are thrown into a convolution with a set of trainable filters to plot out the attributes.

Convolution, activation, pooling, and fully interconnected layers are all part of the basic structure of CNN as depicted in Figure 9. The proposed CNN model has four main layers and an output layer; each of these layers is composed of three convolution layers with a kernel size of three for the first two convolution layers and a kernel size of five for the final convolution layer. Stride equal to one for the first two convolution layers and stride equal to two for the final convolution layer are used; the relu activation function is used for all layers; and, finally, there are three maxpooling layers with a pool size of three and a stride of one. The convolution layer acts as a “filter”, taking the observed pixel values from the input image and transforming them into a single value using the convolution process. When the convolution layer is applied, the original images will be reduced to a smaller matrix. In order to improve the filtered images, backpropagation training will be used. Down-sampling and shrinking the matrix size will help speed up training, as the pooling layer has this purpose. After that, the classification results are output by the completely linked layer (a typical multilayer perceptron).

##### Model Training Using Modified Resnet-50

The Fundamental architecture of the proposed system is founded on the Resnet-50 Model. DL models must account for a staggering amount of structures and hyperparameters (e.g., number of frozen layers, batch size, epochs, and learning rate, etc.). The effect of numerous hyperparameter settings on system functioning is investigated. The Resnet-50 [48] model is updated in this part to serve as a basis for a possible solution. A novel residual learning attribute of the CNN design was created in 2015 by He K. et al. [48]. A standard layer compensates for the residual unit with a missing connection. By connecting to the layer’s output, the skip connection makes it possible for an input signal to pass throughout the network.

A 152-layer model was created due to the residual units, which achieved the 2015 LSVRC2015 competition. Its new residual architecture makes gradient flow and training easier and more efficient thanks to its novel residual structure. A mistake rate of less than 3.6 percent is among the best in the industry. Other variants of ResNet have 34, 50, or 101 layers. Figure 10 shows the original Resnet-50 Model and its modified variants, which we analyze in this study. Figure 10a shows the initial Resnet-50 Model.

Figure 10b demonstrates how the proposed two versions are built: We add a fully connected (FC) layer and two more FC layers, replacing the existing FC and softmax layers in both versions. This Model’s first two layers were trained using the ImageNet dataset [49]. That is why at the beginning the additional layers’ weights will be chosen at random. The weights of all models are then updated using backpropagation, the key algorithm for training neural network architecture. Figure 10b shows how Resnet-50’s initial FC layer, which was already deleted and substituted by the new FC layer of size 512, was swapped with another FC layer of size three and a new softmax layer that was replaced with a novel softmax layer (Figure 10b). The system needs more FC layers for tiny datasets than that for larger ones [50,51].

In the completely linked layer, all neurons are coupled to all other neurons in the layer above and below it. Grading is determined by an activation function that accepts the output from the final FC layer. One of the most popular classifiers in DNN is Softmax, which uses its equations to calculate the probability dissemination of the n output sets. Only the high computational cost of adding a single FC layer prevents this approach from being widely adopted.

One thousand twenty-four bytes make up the first FC layer; three bytes make up the third FC layer. We employ batch normalization to combat network overfitting; this takes place when a model does a great job of retaining information from its training data but lacks the ability to transfer that knowledge to novel testing data. To put it another way, this problem is more likely to arise when the training dataset is small. To account for the inherent unpredictability of the algorithm’s numerous phases, deep neural networks (DNNs) always produce somewhat variable results [52]. Ensemble learning can be used to maximize the functioning of DNN algorithms. We present the “many-runs ensemble” as a means to achieve stacking generalization through numerous training iterations of the same framework.

## 4. Experimental Results

### 4.1. Training and Configuration of Resnet-50 and the Proposed CNN

DL systems have been tested on the HAM10000 dataset to see how well they work and to see how they compare to current best practices. There are two groups of data: 90% for training (9016 images) and 10% for testing (984 images). A total of 10% of the training set is utilized for validation (992). All images were scaled to 227 × 227 × 3 and increased to 39,430 images in the training method. Linux PC with RTX3060 and 8 GB of RAM were used to test the TensorFlow Keras application on this machine. A random image set of 80 percent served as the basis of the suggested DL systems’ training. After training, 10 percent of training data was used as a validation set in which the most accurate weight combinations were saved for future use. The HAM10000 dataset is used to pre-train the suggested framework, which employs the Adam optimizer and a learning rate technique that slows down learning when it becomes stagnant for a prolonged span of time (i.e., validation patience). The subsequent hyperparameters were fed into the Adam optimizer during the training process: Batch sizes range from 2 to 64 with a move of two times the former value; epochs are 50; patience is 10; and momentum is 0.9 for this simulation. An infection form dissemination approach known as “batching” rounds out our arsenal of anti-infective measures.

### 4.2. Set of Criteria for Evaluation

This study section provides an in-depth description of the evaluation metrics and their results. A popular metric for gauging classification efficiency is classifier accuracy (Ac). The number of instances (images) correctly classified divided by the dataset’s total number of examples (images) is the equation’s definition (1). When analyzing the efficiency of image categorization algorithms, precision (Pre) and recall (Rec) are the two most commonly utilized criteria. The greater the number of accurately labeled images, the greater the degree of precision in the Equation (2). The ratio of photographs in the database was successfully categorized to those associated numerically in Equation (3). Having a higher F-score indicates that the system is better at forecasting the future than if it has a lower one. A system’s effectiveness cannot be measured solely on the basis of accuracy or recall. Explanation (4) shows how to calculate the F-score mathematically (Fsc). The last metric is Top N accuracy, where the model N’s highest probability answers must match the expected softmax distribution to be considered “top N accuracy”. A classification is considered correct if at least one of the N predictions matches the label being sought.
(1)$$\mathrm{Ac}=\frac{T^{p}+{\;T}^{n}}{T^{p}+{\;T}^{n}+{\;F}^{p\;}+{\;F}^{n}}$$
(2)$$\mathrm{Pre}=\frac{T^{p}}{T^{p}+{\;F}^{p\;}}$$
(3)$$\mathrm{Rec}=\frac{T^{p}}{T^{p}+{\;F}^{n\;}}$$
(4)$$\mathrm{Fsc}=2*\left(\frac{\mathrm{Pre}*\mathrm{Rec}}{\mathrm{Pre}\;+\;\mathrm{Rec}}\right)$$

True positives are denoted by the symbol (T^p^) and are positive cases that were successfully predicted, while true negatives (T^n^) are negative situations that were accurately predicted. False positives (F^p^) are positive situations that were mistakenly predicted, and false negatives (F^n^) are negative examples that were wrongly forecasted.

### 4.3. Performance of Various DCNN Models

Data from the HAM10000 skin lesion categorization challenge dataset are being used to train and evaluate a variety of DCNNs (including CNN and Resnet-50). The results of multiple assessments of the HAM100000 dataset for the suggested systems are shown using a 90–10 split between training and testing. In order to minimize the time it takes to complete the project, this division was decided. Models were trained for 50 epochs using 10% of the training set as a validation set, a batch size of 2 to 64, and learning rates ranging from 1 × 10^4^, 1 × 10^5^, and 1 × 10^6^ for CNN and Resnet-50, respectively. Resnet-50 was further fine-tuned by freezing varying numbers of layers to reach the useful accuracy possible. A model ensemble was created by running a number of runs on the same Model with the same parameters. Because the weights are created randomly for each run, the accuracy varies from run to run. Only the highest run outcome is stored and illustrated in Table 3 and Table 4, one for CNN and one for Resnet-50 training on HAM10000 datasets, respectively. It demonstrates that the best result obtained using CNN and Resnet-50 is 86% and 85.3%, respectively. Figure 11 and Figure 12 demonstrate the confusion matrix using CNN and Resnet-50, respectively. By applying the proposed approach to the test set, the confusion matrix was obtained. According to the confusion matrix, the suggested technique can identify nv lesions with 97% accuracy (770 correctly classified image out of 790 total images), which is extremely desirable for real-world applications using the CNN model and 94% (749 correctly classified image out of 790 total images) using the modified version of Resnet-50.

Figure 13 shows two successful examples of classifying two images, one belonging to the Nv class and the other to the Akiec class. A total number of images used for each class in the Ham10000 dataset is shown in Table 5 and Table 6, which reveal the number of images used for testing in each class. According to the results, it is obvious that the Nv class has the biggest number of images with 795; its pre, rec, and Fsc are all very high, equal to 91 percent, 97 percent, and 94 percent, respectively, using the CNN model. The values of these parameters are 94 percent, 94 percent, and 94 percent, respectively, using the modified Resnet-50 model.

Using lesion images to help dermatologists diagnose infections more accurately and reduce their workload has now been proven feasible in real-world settings.

### 4.4. Evaluation with Other Methods

Efficacy to that of other approaches is conducted. Table 7 indicates that our technique outperforms other approaches in terms of efficiency and effectiveness. Overall, the proposed inception model achieves an 86 percent accuracy rate, transcending the current methods.

### 4.5. Discussion

As we discovered, other methods could not meet our degree of accuracy. One of three contributing elements is the general resolution enhancement of ESRGAN, which we believe is responsible for this. In addition, we deploy a variety of architectures, each with a varied ability to generalize and adapt to diverse types of data. Transfer learning architectures could not classify medical images more accurately due to a lack of distinctive features. Resnet-50’s classification accuracy was worse than the proposed CNN when applied to medical images, even though it was better at identifying natural images. More generalizable qualities of CNN’s shorter networks suggest that they can be used for a wider range of images. Deeper networks such as Resnet-50, on the other hand, can learn abstract properties that can be used in any sector. CNN features are more generalizable and adaptable for medical imaging because they lack semantic relevance to natural images (compared to Resnet-50). Fine-tuning networks, in turn, made the two models more accurate. CNN’s accuracy improved the greatest compared to Resnet-50. Deep networks, as opposed to shallow ones, were found to be more likely to pick up significant information when trained on a smaller dataset. Shown in Figure 11 and Figure 12 are the results of the indicated processes, which were adequate. Table 7 displays that ResNet and CNN, in references [55,56], yielded 77% and 78% accuracy, respectively. Researchers evaluated the accuracy of their model against the results of these two research projects that used the same dataset and trained their models using the same methods (Convolutional Neural Networks and Resnet-50) so that comparisons could be made easily. To witness its robustness further, the proposed CNN model outperforms two other referenced works [14,57] while being trained on a significantly smaller dataset (9016 images vs. 100,000 in the ImageNet Dataset).

## 5. Conclusions

Researchers devised a method for promptly and accurately diagnosing seven different types of cancer by analyzing skin lesions. The suggested method uses image-enhancing techniques to brighten the lesion image and remove noise. Preprocessed lesion medical imaging was used to train CNN and modified Resnet-50 to avoid overfitting and to boost the overall competence of the suggested DL approaches. The proposed approach was challenged using a dataset of lesion images known as the HAM10000 dataset. When employing CNN and a modified Resnet-50, the conception model had an accuracy rate of 85.3 percent and 85.98 percent ≈ 86 percent, respectively, comparable to the accuracy rate of professional dermatologists, as proposed. In addition, the research’s originality and contribution lie in its use of ESRGAN as a pre-processing step with the various models (designed CNN and modified Resnet50) and in its contribution to the field. Compared to the pre-trained Model, our new Model performs similarly. Current models are outperformed by the proposed system, as demonstrated by comparison studies. Experiments on a big and complicated dataset, including future cancer cases, are required to demonstrate the efficacy of the suggested method. In the future, DenseNet, VGG, or AlexNet may be utilized to evaluate the cancer dataset. Lesion-less skin and lesioned skin are not always caused by skin cancer; it may also be a confounding factor in clinical diagnosis. In future, we will add this into the dataset to test the effectiveness of the model further.

## Figures and Tables

**Figure 1 healthcare-10-02481-f001:**
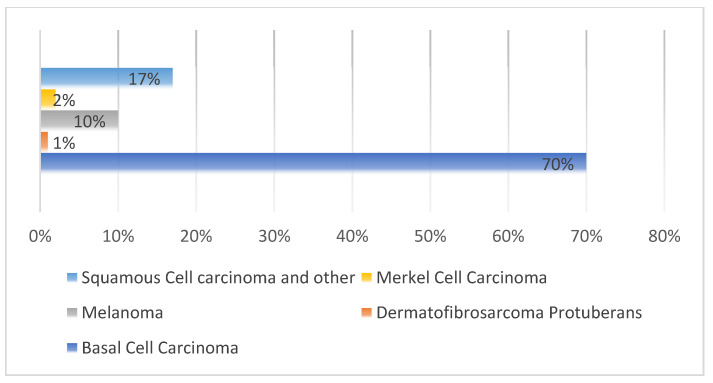
According to Reference [4], a number of different types of skin cancer are widespread.

**Figure 2 healthcare-10-02481-f002:**
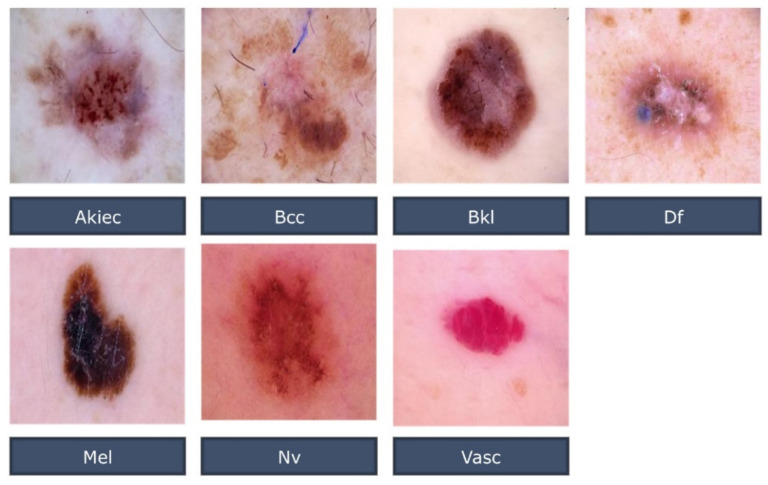
Examples of HAM10000 Dataset.

**Figure 3 healthcare-10-02481-f003:**
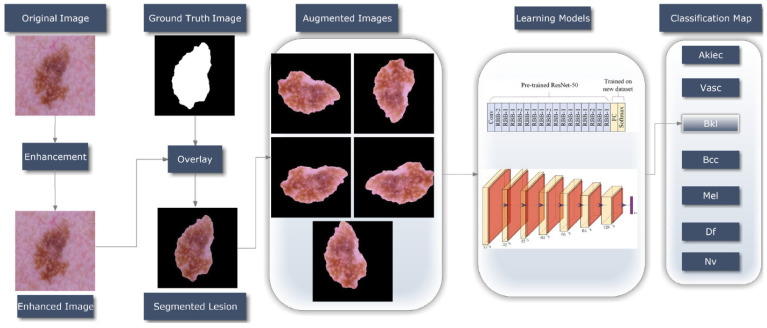
An overall process of how to recognize skin cancer.

**Figure 4 healthcare-10-02481-f004:**
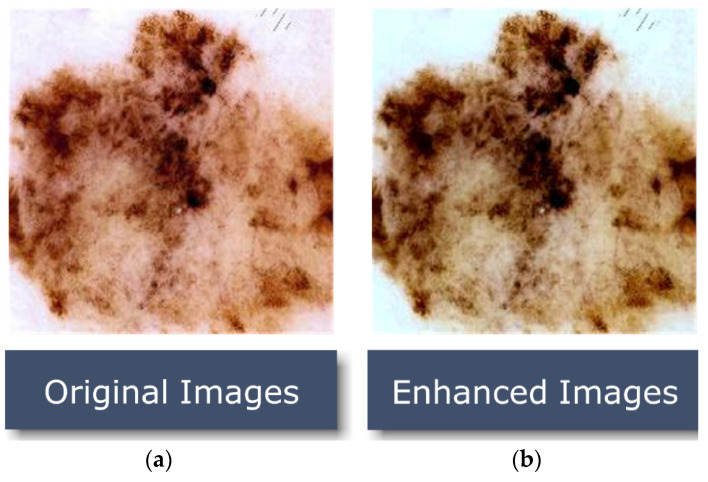
Proposed image-enhancement algorithm results; (**a**) image in its raw form; (**b**) an enhanced version of that image.

**Figure 5 healthcare-10-02481-f005:**
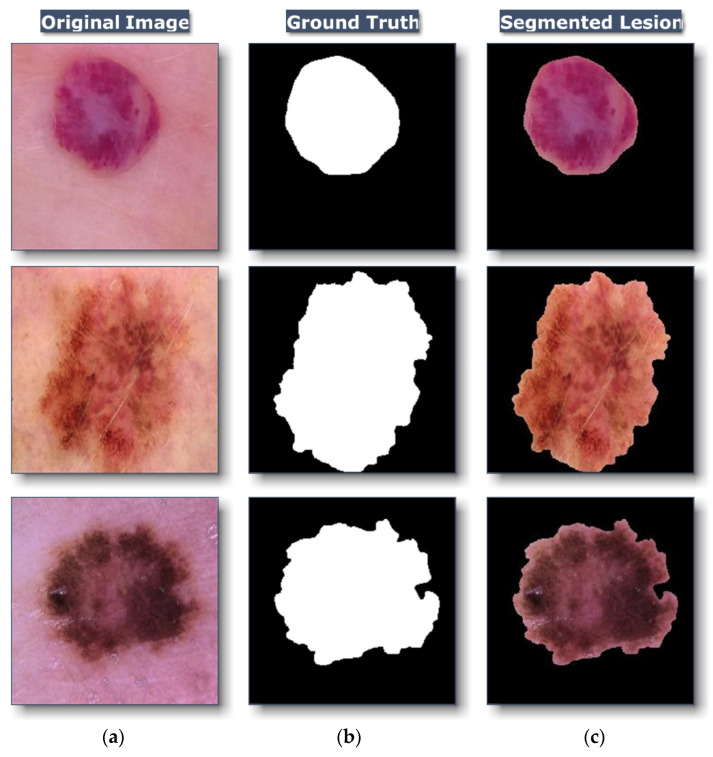
Samples of (**a**) original Image, (**b**) ground truth, and (**c**) the segmented ROI.

**Figure 6 healthcare-10-02481-f006:**
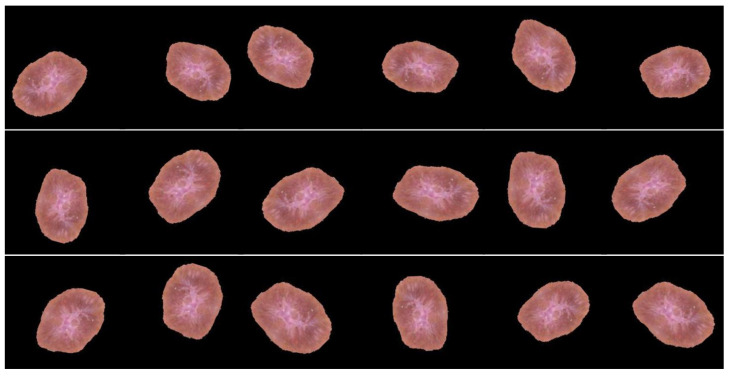
Samples of image augmentation for the same image.

**Figure 7 healthcare-10-02481-f007:**
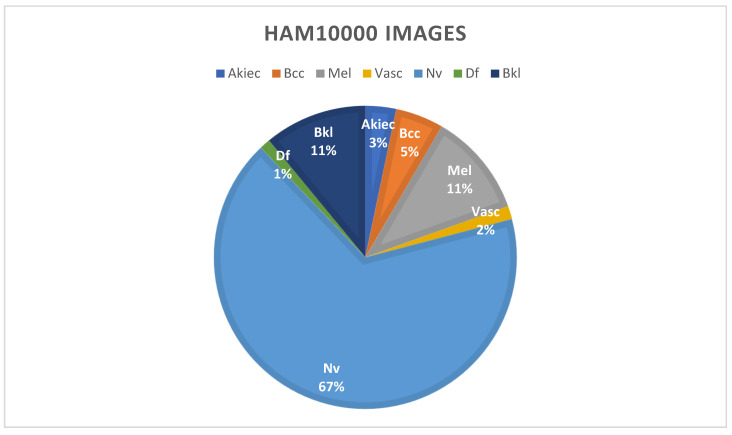
Unbalanced dataset before applying augmentation techniques.

**Figure 8 healthcare-10-02481-f008:**
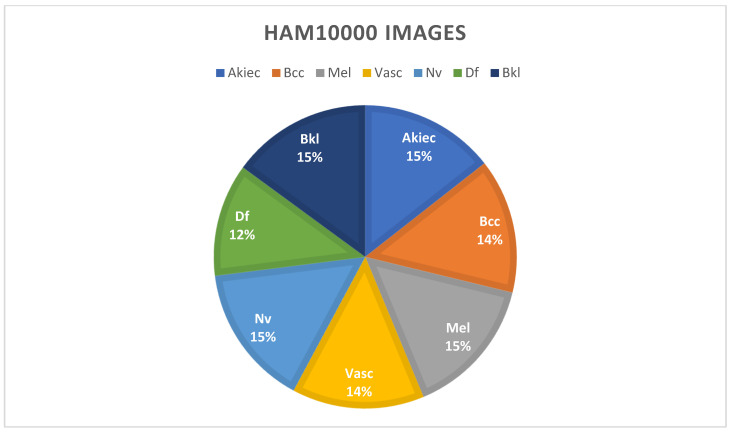
Balanced dataset after applying augmentation techniques.

**Figure 9 healthcare-10-02481-f009:**
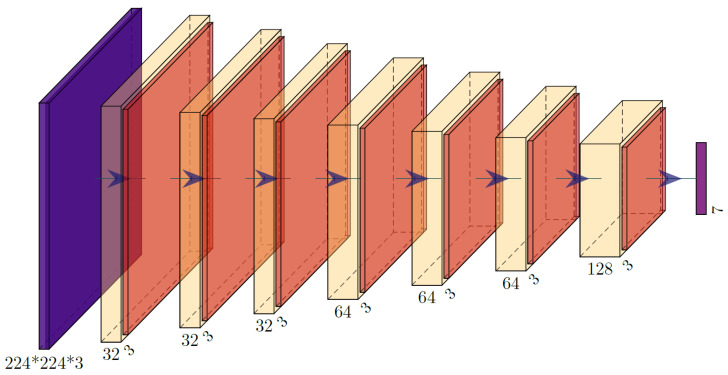
Proposed CNN architecture.

**Figure 10 healthcare-10-02481-f010:**
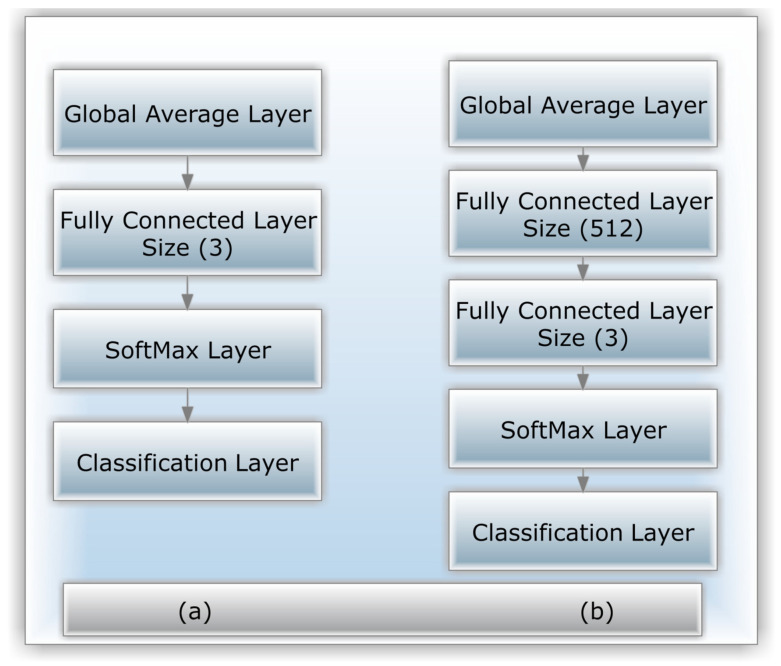
Versions of the Resnet-50 model that were modified; (**a**) the initial pre-trained model; (**b**) the addition of one FC.

**Figure 11 healthcare-10-02481-f011:**
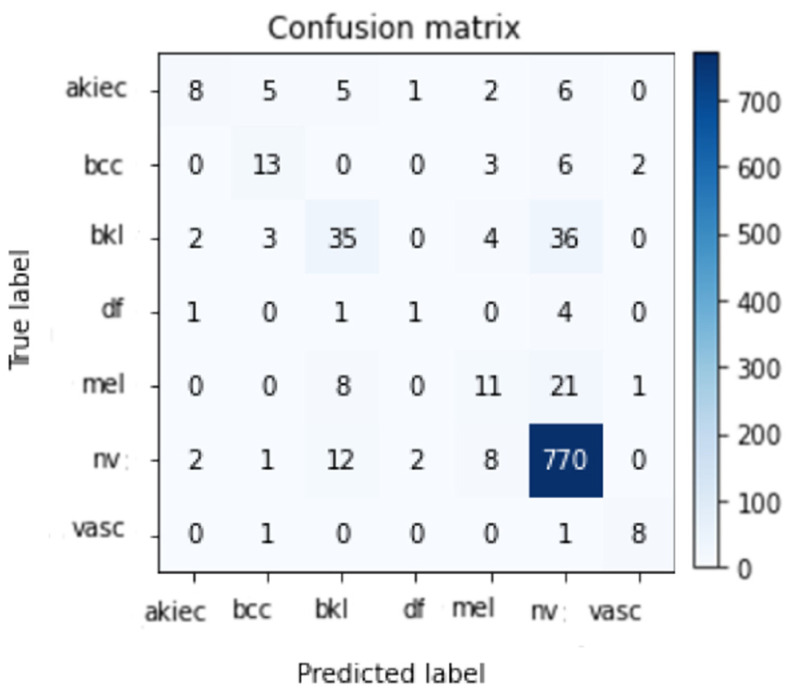
Best confusion matrix of CNN.

**Figure 12 healthcare-10-02481-f012:**
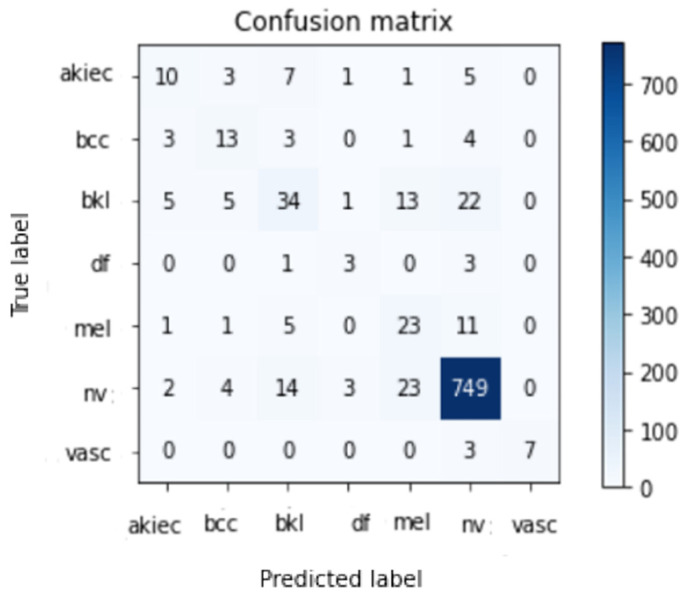
Best confusion matrix of Resnet-50.

**Figure 13 healthcare-10-02481-f013:**
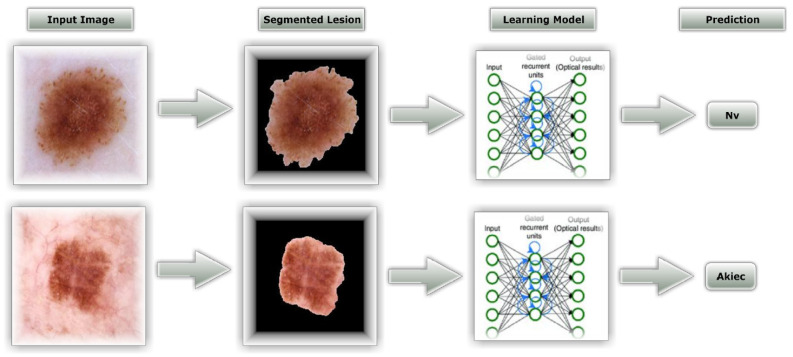
Example of testing classification phase.

**Table 1 healthcare-10-02481-t001:** Methods, data, and results for skin cancer detection available nowadays.

Notable Designs	Size	Dataset	Methods	# Classes
[3]	2298	PAD-UFES-20	EfficientNetB3 + Extreme Gradient Boosting (XGB)	6
[19]	1600	ISIC-2017	swarm intelligence (SI)	2
	1600	ISIC-2018		
	1000	PH-2		
[33]	300	HAM10000	CNN + XGBoost	5
[34]	1323	HAM10000	InSiNet	2
[35]	7470	HAM10000	ResNet50	7
[36]	1000	ISIC	RF + Support Vector Machine (SVM)	8
[37]	6705	HAM10000	CNN	2
[38]	10,015	HAM10000	AlexNet	7
[39]	10,015	HAM10000	CNN	7
[40]	4753	Atlas	ResNet-152	12
[41]	10,015	HAM10000	MASK-RCNN	7
[42]	10,015	HAM10000	DenseNet121	7

**Table 2 healthcare-10-02481-t002:** A balanced dataset resulting from applying Augmentation (oversampling) techniques. As part of the data expansion, segmented photos were included.

Class	Number of Training Images
Akiec	5684
Bcc	5668
Mel	5886
Vasc	5570
Nv	5979
Df	4747
Bkl	5896
Total	39,430

**Table 3 healthcare-10-02481-t003:** Best accuracy using CNN learning model.

Acc	Top-2 Accuracy	Top-3 Accuracy	Pre	Rec	Fsc
0.8598	0.9400	0.9726	0.84	0.86	0.8598

**Table 4 healthcare-10-02481-t004:** Best accuracy after fine-tuning using modified Resnet-50 transfer learning model.

Acc	Top-2 Accuracy	Top-3 Accuracy	Pre	Rec	Fsc
0.8526	0.9329	0.9695	0.86	0.85	0.8526

**Table 5 healthcare-10-02481-t005:** Detailed results for each class using CNN learning model.

	Pre	Rec	Fsc	Total Images
Akiec	0.62	0.30	0.4	27
Bcc	0.57	0.54	0.55	24
Bkl	0.57	0.44	0.50	80
Df	0.25	0.14	0.18	7
Mel	0.39	0.27	0.32	41
Nv	0.91	0.97	0.94	795
Vasc	0.73	0.80	0.76	10
Average	0.84	0.86	0.86	984

**Table 6 healthcare-10-02481-t006:** Detailed results for each class using modified Resnet-50 learning model.

	Pre	Rec	Fsc	Total Images
Akiec	0.48	0.37	0.42	27
Bcc	0.48	0.54	0.51	24
Bkl	0.55	0.42	0.48	80
Df	0.38	0.43	0.40	7
Mel	0.37	0.59	0.45	41
Nv	0.94	0.94	0.94	795
Vasc	1.00	0.70	0.82	10
Average	0.86	0.85	0.85	984

**Table 7 healthcare-10-02481-t007:** Comparison with other methods

Reference	Dataset	Model	Accuracy
[14]	HAM10000	RegNetY-3.2GF	85.8%
[49]	HAM10000	AlexNet	84%
[50]	HAM10000	MobileNet	83.9%
[51]	ISIC2018	CNN	83.1%
[51]	ISIC2018	Resnet-50	83.6%
[52]	HAM10000	MobileNet, VGG-16	80.61%
[53]	ISIC2018	Resnet-50	85%
[54]	HAM10000	Support Vector Machine (SVM), Logistic Regression (LR), Random Forest (RF), AdaBoost (Adaptive Boosting), Balanced Bagging (BB) and Balanced Random Forest (BRF)	74.75%
[55]	HAM10000	CNN	77%
[56]	HAM10000	ResNet, Xception, and DenseNet	78%, 82%, 82%
[57]	HAM10000	MobileNet and LSTM	85%
Proposed	HAM10000	CNN	86%
Proposed	HAM10000	Modified Resnet-50	85.3%

## Data Availability

Publicly available datasets were analyzed in this study. These data can be found here: (https://www.kaggle.com/datasets/kmader/skin-cancer-mnist-ham10000?select=HAM10000_metadata.csv (accessed on 13 April 2022)).

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
