# Peer review of "Melanoma Detection Using Deep Learning-Based Classifications"

_healthcare, 2022, doi:10.3390/healthcare10122481_

Round 1

Reviewer 1 Report

This study provides an efficient deep learning method for accurate extraction of skin cancer lesions. The data is first augmented using trained ESRGAN, then the target region is segmented using segmentation techniques, and finally the images are classified using CNN and a modified version of Resnet-50. However, there are a few major changes must be made before reconsideration:

1.      Table 7 lists many related studies. However, under the same dataset HAM10000, the accuracy of the models presented in this study (CNN 86%, Modified Resnet-50 85.3%) is not significantly better than RegNetY-3.2GF (85.8%) and MobileNet and LSTM (85%). The authors need to justify why their models are better than the old ones through other evaluation metrics.

2.      In the conclusion section, it is mentioned that the originality of the study uses ESRGAN to preprocess the graphs to improve the resolution. However, this study does not have a control group: what will happen if ESRGAN is not used? The authors must do such a comparison to justify their conclusion.

3.      Lesion less skin and lesioned skin not caused by skin cancer may also be a confounding factor in clinical diagnosis, and if available, this part of the data needs to be added to the dataset to further test the effectiveness of the model.

4.      Line 389: the confusion matrices needs more explanation: what can the confusion matrices tell us? Since the “nv” shows a significantly darker color than others, any information can be inferred?

5.      The authors need to work on their figures with more efforts. For example, the confusion matrices do not have names for x-axis and y-axis. Which one is predicted value and which one is experimental value? Also, the capitalization of the first letter of figure captions are inconsistent (e.g. capitalized in Figure 12 but not in Figure 13). Also, the authors must provide detail explanation of the figures in the figure captions, if they have not explained the figure in detail in the main text.

6.      The layout of this paper needs to be significantly improved. I already mentioned to the editor that the original version of the paper has figure number restarted in the middle of the paper (the editor fixed the problem for you which should have been the authors’ duty!) Also, there are a lot of excessive blanks in the main text (e.g. line 105 before “DL”), in figure captions (Line 409), and also inconsistent line spaces (e.g. line 73 to 89).

7.      Why there is a list of authors’ names at the end of the abstract?

8.      What’s going on with Figure 10? It is not even a complete picture!

Again, as I mentioned to the editor previously, I was shocked to see the poor quality of the layout of this paper. If this problem is not solved in the revision, I will definitely recommend to reject this paper.

Author Response

This study provides an efficient deep learning method for accurate extraction of skin cancer lesions. The data is first augmented using trained ESRGAN, then the target region is segmented using segmentation techniques, and finally the images are classified using CNN and a modified version of Resnet-50. However, there are a few major changes must be made before reconsideration:

Comment 1: Table 7 lists many related studies. However, under the same dataset HAM10000, the accuracy of the models presented in this study (CNN 86%, Modified Resnet-50 85.3%) is not significantly better than RegNetY-3.2GF (85.8%) and MobileNet and LSTM (85%). The authors need to justify why their models are better than the old ones through other evaluation metrics.

Author response:  Thank you for your comment.

Author action: All transfer learning models have been trained on the ImageNet Dataset, which contains 100,000 images, whereas our CNN model has been trained on only 9016 images, and it offers result better than the two stated works, which supports the robustness of our provided CNN.

Comment 2: In the conclusion section, it is mentioned that the originality of the study uses ESRGAN to preprocess the graphs to improve the resolution. However, this study does not have a control group: what will happen if ESRGAN is not used? The authors must do such a comparison to justify their conclusion.

Author response:  Thank you for your comment.

Author action: Table 7 shows that both references [59], which used ResNet, and references [60], which utilized CNN, both obtained results of 77% and 78% respectively.

To demonstrate the robustness of our model, we compared it to these two different studies that used the same dataset; both of these studies used CNN and Resnet-50 for training, which facilitated the comparison process; the only difference between our study and these other studies was the addition of image enhancement (ESRGAN), which resulted in a more accurate model.

Comment 3: Lesion less skin and lesioned skin not caused by skin cancer may also be a confounding factor in clinical diagnosis, and if available, this part of the data needs to be added to the dataset to further test the effectiveness of the model.

Author response:  Thank you for your comment.

Author action: We are really thankful for the suggested change, however, changing dataset at this stage needs the repetition of the whole experiment process. therefore, we have added this suggestion in our future work.

The following text is added in the paper in the conclusion section in red as follows:

Lesion-less skin and lesioned skin are not always caused by skin cancer, it may also be a confounding factor in clinical diagnosis. In future, we will add this into the dataset to further test the effectiveness of the model.

Comment 4: Line 389: the confusion matrices needs more explanation: what can the confusion matrices tell us? Since the “nv” shows a significantly darker color than others, any information can be inferred?

Author response:  Thank you for your comment.

Author action: We have updated the two figures used to illustrate the results of confusion matrix 

Figure 11. Best confusion matrix of CNN.

Figure 12. Best confusion matrix of Resnet-50.

We also updated the text in red as follows:

By applying the proposed approach to the test set, the confusion matrix was obtained. According to the confusion matrix, the suggested technique can identify nv lesions with 97% accuracy (770 correctly classified image out of 790 total images), which is extremely desirable for real-world applications using CNN model and 94% (749 correctly classified image out of 790 total images), using the modified version of Resnet-50.

Comment 5: “The authors need to work on their figures with more efforts. For example, the confusion matrices do not have names for x-axis and y-axis. Which one is predicted value and which one is experimental value? Also, the capitalization of the first letter of figure captions are inconsistent (e.g. capitalized in Figure 12 but not in Figure 13). Also, the authors must provide detail explanation of the figures in the figure captions, if they have not explained the figure in detail in the main text.

Author response:  Thank you for your comment.

Author action: We have updated figure 11 and figure 12

Figure 11. Best confusion matrix of CNN.

Figure 12. Best confusion matrix of Resnet-50.

We also updated the text in red as follows:

By applying the proposed approach to the test set, the confusion matrix was obtained. According to the confusion matrix, the suggested technique can identify nv lesions with 97% accuracy (770 correctly classified image out of 790 total images), which is extremely desirable for real-world applications using CNN model and 94% (749 correctly classified image out of 790 total images), using the modified version of Resnet-50.

Comment 6: The layout of this paper needs to be significantly improved. I already mentioned to the editor that the original version of the paper has figure number restarted in the middle of the paper (the editor fixed the problem for you which should have been the authors’ duty!) Also, there are a lot of excessive blanks in the main text (e.g. line 105 before “DL”), in figure captions (Line 409), and also inconsistent line spaces (e.g. line 73 to 89).

Author response:  Thank you for your comment.

Author action: We've taken note, and we do our best to address this concern as shown in the attached file.

Comment 7: Why there is a list of authors’ names at the end of the abstract?

Author response:  Thank you for your comment.

Author action: We've taken note, and we updated this mistake

Comment 8:  What’s going on with Figure 10? It is not even a complete picture!

Author response:  Thank you for your comment.

Author action: figure 10 shows only the last updated layers of the modified version of ResNet-50

Reviewer 2 Report

1. The description of segmentation masks (ground truth) is missing. Are they included in the HAM10000 database? Or were they made for the purpose of this study? If so, who made them and how.

2. The necessity of segmentation as image pre-processing severely limits the possibilities of using the proposed algorithm. How to test a new image that does not come from the database and does not have a mask image? Is the ground truth mask calculated automatically and how?

3. In the proposed solution, two mechanisms were used to improve training compared to standard approaches based on deep learning: enhancement and segmentation of the input image.

The authors say that the high quality of the classification is mainly due to the use of ESRGAN preprocessing, but this has not been proven. Separate comparisons of the classification results for the enhancement only and the segmentation itself should be made.

It seems to me that getting rid of unnecessary skin lesion background data can significantly improve the classification while introducing a limitation in the use of the algorithm in practice.

I believe that the statement in the Conclusion (line 444-446): "In addition, the research's originality and contribution lie in its use of ESRGAN as a pre-processing step with the various models (Designed CNN and modified Resnet50) and in its contribution to the field." is not proven in paper.

4. Figure 4 shows the original and the enhanced image. They look the same. Rather, an image should be generated that shows the differences between the two images.

Author Response

Comment 1: The description of segmentation masks (ground truth) is missing. Are they included in the HAM10000 database? Or were they made for the purpose of this study? If so, who made them and how.

Author response:  Thank you for your comment.

Author action: The ground truth images are provided with HAM10000 dataset, and it has been mentioned in the text in red as follows:

Following the protocol for preparing images, ROI from the dermoscopy image is segmented. To generate ROI in each image, a ground truth mask ,which have been provided by the HAM10000 dataset for general purpose usage, would be applied to the enhanced image., as demonstrated in figure 5.

Comment 2: The necessity of segmentation as image pre-processing severely limits the possibilities of using the proposed algorithm. How to test a new image that does not come from the database and does not have a mask image? Is the ground truth mask calculated automatically and how?

Author response:  Thank you for your comment.

Author action: We've taken note, and we do our best to address this concern in the future work, as this work only test the model on the available test images in the dataset.

Comment 3: In the proposed solution, two mechanisms were used to improve training compared to standard approaches based on deep learning: enhancement and segmentation of the input image.

The authors say that the high quality of the classification is mainly due to the use of ESRGAN preprocessing, but this has not been proven. Separate comparisons of the classification results for the enhancement only and the segmentation itself should be made.

It seems to me that getting rid of unnecessary skin lesion background data can significantly improve the classification while introducing a limitation in the use of the algorithm in practice.

I believe that the statement in the Conclusion (line 444-446): "In addition, the research's originality and contribution lie in its use of ESRGAN as a pre-processing step with the various models (Designed CNN and modified Resnet50) and in its contribution to the field." is not proven in paper.

Author response:  Thank you for your comment.

Author action: Table 7 shows that both references [59], which used ResNet, and references [60], which utilized CNN, both obtained results of 77% and 78% respectively.

To demonstrate the robustness of our model, we compared it to these two different studies that used the same dataset; both of these studies used CNN and Resnet-50 for training, which facilitated the comparison process; the only difference between our study and these other studies was the addition of image enhancement (ESRGAN), which resulted in a more accurate model.

Comment 4: Figure 4 shows the original and the enhanced image. They look the same. Rather, an image should be generated that shows the differences between the two images.

Author response:  Thank you for your comment.

Author action: We have revised Figure 4 and updated it with a new image, as shown in the figure

The quality of the enhanced image has been improved 

Round 2

Reviewer 1 Report

Two minor issues to address:

1. The authors did not fix Figure 10 as I suggested in old comment 8. Figure 10 is still inappropriate for publication. The authors should draw a flowchart by themselves instead of providing a screenshot containing incomplete words such as "Classification..." and "classificationLa...". Some boundaries of the boxes are also missing. 

2. Regarding my old comment 1 and 2, they need to add their explanations to these problems in the main text, not just answering my question. I asked these question because I could not find the answer in their manuscript. 

Author Response

This study provides an efficient deep learning method for accurate extraction of skin cancer lesions. The data is first augmented using trained ESRGAN, then the target region is segmented using segmentation techniques, and finally the images are classified using CNN and a modified version of Resnet-50. However, there are a few major changes must be made before reconsideration:

Comment 1:  The authors did not fix Figure 10 as I suggested in old comment 8. Figure 10 is still inappropriate for publication. The authors should draw a flowchart by themselves instead of providing a screenshot containing incomplete words such as "Classification..." and "classificationLa...". Some boundaries of the boxes are also missing.

Author response:  Thank you for your comment.

Author action: We have revised Figure 10 and updated it with a new image, as shown in the updated manuscript

Comment 2: Table 7 lists many related studies. However, under the same dataset HAM10000, the accuracy of the models presented in this study (CNN 86%, Modified Resnet-50 85.3%) is not significantly better than RegNetY-3.2GF (85.8%) and MobileNet and LSTM (85%). The authors need to justify why their models are better than the old ones through other evaluation metrics.

Author response:  Thank you for your comment.

Author action:

Regarding old Comment 1 and Comment 2, the following text has been added to the discussion section in Red as follows:

Table 7 displays that ResNet and CNN, in References [59,60], yielded 77% and 78% accuracy, respectively. Researchers evaluated the accuracy of their model against the results of these two research that used the same dataset and trained their models using the same methods (Convolutional Neural Networks and Resnet-50), so that comparisons could be made easily. To further witness to its robustness, the proposed CNN model outperforms two other referenced works [14,60] while being trained on a significantly smaller dataset (9,016 images vs. 100,000 in the ImageNet Dataset).

Reviewer 2 Report

I still cannot see the difference between images in Fig. 4. Please replace it with the differential image.

Author Response

Comment 1: I still cannot see the difference between images in Fig. 4. Please replace it with the differential image.

Author response:  Thank you for your comment.

Author action: We have revised Figure 4 and updated it with a new image, please see attached updated manuscript

The quality of the enhanced image has been improved